# Application of In-Paddock Technologies to Monitor Individual Self-Fed Supplement Intake and Liveweight in Beef Cattle

**DOI:** 10.3390/ani10010093

**Published:** 2020-01-06

**Authors:** José A. Imaz, Sergio García, Luciano A. González

**Affiliations:** 1School of Life and Environmental Sciences, Faculty of Science, The University of Sydney, Sydney, NSW 2570, Australia; sergio.garcia@sydney.edu.au (S.G.); luciano.gonzalez@sydney.edu.au (L.A.G.); 2Instituto Nacional de Tecnología Agropecuaria (INTA), Capital Federal 1033, Argentina; 3Sydney Institute of Agriculture, The University of Sydney, Sydney, NSW 2570, Australia

**Keywords:** technologies, individual, supplement intake, liveweight, monitoring, automate

## Abstract

**Simple Summary:**

Individual, daily and simultaneous measures of key variables to manage cattle have been traditionally difficult to achieve in grazing animals. However, nowadays, this could be achieved using technologies which are placed ‘in paddock’ such as automated weighing scales to measure liveweight (LW) and electronic feeders (EF) to measure supplement intake. We used both technologies to study the interplay between the intake of a self-fed supplement (molasses-lick blocks, MLB), growth, and feeding behavior of individual animals fed a sequence of different feed types. We identified a large individual variability in MLB intake with some animals consuming supplement regularly while others not consuming supplement at all. Regular consumers tended to grow more rapidly. Additionally, our results indicate that animals’ MLB intake can be predicted using the number of visits to the EF and their duration. In-paddock technologies could aid to quantify key factors, such as individual variability of supplement intake and LW, that would otherwise remain undetected.

**Abstract:**

The aim of this study was to assess the ability of in-paddock technologies to capture individual variability of self-fed supplement intake (molasses-lick blocks, MLB), feeding behavior, and liveweight (LW) in grazing beef cattle. An electronic feeder (EF) and in-paddock walk-over-weighing system (WOW) were installed to measure, daily and simultaneously, individual MLB intake and LW. Cattle grazed (pastures and oat crops) and were fed (lucerne and oaten hay) during a 220 day trial. Over the entire period, we were able to quantify a large variability in MLB intake between individuals (*p* < 0.01; ranging from 0 to 194 g/hd per day). Liveweight change (*p* < 0.05, R = 0.44) and feeding behaviour (e.g., feeding frequency and duration, *p <* 0.01; R^2^ > 0.86) were positively correlated with MLB intake over the entire period but these correlations seemed to be affected by the type of feed. The intake of MLB seems to be explained by the individual behaviour of animals rather than the entire group. The use of in-paddock technologies enabled remote measurement of variability in supplement intake and cattle growth. The ability to monitor LW and feeding behavior of individual animals in a group could allow automatic individualized feeding of grazing cattle (amount and type of supplement) and managing low-performing animals under grazing conditions.

## 1. Introduction

Managing the productivity of supplemented grazing beef cattle has been challenging. This is mostly because of the large variability in liveweight (LW) and supplement intake that exists between and within individual animals [1,2]. However, labour and time constraints have prevented the frequent measurement (e.g., daily, weekly) of these variables on individual animals without altering their normal feeding behaviour and welfare [3,4]. As a result, frequency of data collection was, in many cases, lower than that required to precisely describe individual variation of LW and supplement intake which were traditionally measured on group instead of individual basis [5]. The use of ruminal markers is impractical to measure individual supplement intake accurately for periods longer than 2 weeks [6,7]. Furthermore, the lack of individual data restrains correlating feeding and performance variables throughout seasons.

Nowadays, digital technologies such as electronic feeders (EF) and in-paddock walk-over-weighing scales (WOW) could be used to measure individual supplement intake and LW of cattle in near real-time and without human intervention [8]. Electronic feeders can also record feeding behaviour including frequency, duration, and size of every single feeding event [9]. This technology-based approach could enable the study of free choice (self-fed) supplement intake, such as molasses-lick-blocks (MLB). These MLB can provide energy, protein, minerals, medication, and additives while controlling supplement intake through the block hardness [10].

Only a few studies reported on feeding behaviour of self-fed supplements using EF [9,11] whereas no studies were found that reported the response of MLB-supplemented beef cattle fed with a range of forage types over time. Monitoring individual feeding behaviour might have the potential to predict MLB intake when measuring block disappearance is not possible. Integrating data streams obtained from EF and WOW could allow identification of poor performing animals and determine if their lower performance is related, among other factors, to supplement intake.

The lack of individual and simultaneous measures of supplement intake and LW had limited previous research on molasses-based supplements to explore differences between animals [5]. Previous research indicated that individual supplement intake of group-fed cattle can largely exceed the target amount in some animals while others do not consume any supplement [1]. Particularly, studies were not conclusive on whether molasses-based supplementation has positive effects on liveweight change (LWC) or not. Improved LWC was mostly reported while feeding low-quality forages, however, others have not demonstrated any positive effect of MLB in LWC of animals fed either low- or high-quality forages [1,5].

The aims of the present study were to: (a) measure individual intake of supplement (MLB) and liveweight remotely; and (b) integrate MLB intake, feeding behaviour (EF) and liveweight (WOW) data to monitor their relationships. We hypothesized that in-paddock feeding and animal weighing systems could allow measuring the relationship between MLB intake and performance in real-time of individual animals feeding a sequence of feedstuffs for long periods of time.

## 2. Materials and Methods

All experimental procedures were approved by the institutional Animal Ethics Committee from The University of Sydney (Approval 2017/1162).

### 2.1. Experimental Details

A feeding trial was carried out for 220 days (from 16 April to 22 November 2017) using weaned beef cattle (Charolais × Angus crossbred) between 6 to 7 months of age (n = 27, 11 heifers and 16 steers; Initial LW = 190.10 ± 34.08 kg). Cattle were tagged with electronic identification (EID) and grazed rotationally on 24.7 ha divided into 18 paddocks of temperate pastures and oat crops at John Pye Farm (Greendale, NSW, Australia). Hay and concentrate supplementation were introduced due to drought. All animals were allowed to lick a single molasses-lick block inside an EF allowing for ad-libitum consumption during the entire trial period (40 kg block; 4 Season Co. Pty Ltd., Creastmead, QLD, Australia). Further details regarding the composition of pastures and nutritional value of supplements are reported later on this manuscript. The chemical composition of MLB (DM basis) was 8.9% of CP; neutral detergent fibre, NDF: 2.83%; and dry organic matter digestibility, DOMD: 71.5%; and the ingredient composition was 42% molasses, 9% salt, 3% urea, 3% vegetable oil, 1.3% phosphorus, 3% calcium, 4% magnesium, 15% cottonseed meal, 2% lasalocid (Bovatec, Zoetis, Parsippany, NJ, USA), 6% trace mineral mix (copper, cobalt, iodine, and zinc), and 11.7% water. Total rainfall during the trial was 170.2 mm.

### 2.2. In-Paddock Measurements and Yard Setup

A figure showing the layout of the yard setup is given in Imaz et al. [12]. A two-section yard centrally located to the paddocks (15 × 25 m) was built, enclosing the only water point, where both WOW and EF were located. An in-paddock WOW station with an auto drafter gate was placed at the entry of the yard to record LW, EID, date and time (Precision Pastoral Ltd., Alice Spring, Northern Territory, Australia). The WOW consisted of a platform (0.8 m width × 2.4 m length) placed over two load bars and mounted along wooden panels (3 m length × 2 m height) on both sides. Spear gates were used at the entry of the WOW and each exit gate to allow animals to move in only one direction. Animals were trained to use the WOW following the procedure proposed by [13] during the first 4 weeks after weaning and finally moved to the final trial location on 13 March.

An EF was installed, immediately before the exit gate, to record EID, time, date and MLB disappearance (Smartfeed developed by C-lock Inc., Rapid City, SD, USA). Only supplemented animals (n = 27) had access to the EF because animals were drafted (WOW gate) to either the right or left section of the yard depending on the treatment each animal was allocated. Animals that were not supplemented (NS, n = 25) were grazing together with the MLB-supplemented group but drafted to a different yard and results were reported in a companion paper [12]. A single MLB was placed inside the supplemented yard to expose all animals to the supplement from 13 March to 16 April when records started to be recorded. Animals were trained to use the EF leaving the pneumatic gate continuously open for 2 weeks. At week 3, the pneumatic gate was set to close half way (50% closed) and finally set to close fully on week 4. Up to our knowledge, this model of EF was not used before for measuring the individual intake of MLB by cattle. However, the same EF was recently used by Reuter et al. [9] and Williams el at. [11] to measure the individual supplement intake of cattle which ranged from 0 to 2.78 kg/hd per day. In the present study, the EF was calibrated twice a week on average by comparing the weight reported by the system with a known weight (10 kg). In addition, weights that were too high were identified using video cameras and found to be the result of animals hitting the bin resulting in abrupt changes of the weight recorded by the system. Similar observations of animals pushing on the feeder bin were reported by Reuter [9]. In addition, comparisons between MLB disappearance and the total intake calculated as the sum of the intake of all visits by each individual animal were carried out to monitor the accuracy of the system. For example, the total MLB disappearance during a 4-week period (from 1 July to 28 July 2017) was 20.85 kg whereas the calculation of the sum of intake of all feeding events in the same period was 20.30 kg.

### 2.3. Nutritional Management

Animals grazed pasture and forage crops, and fed concentrates depending on forage availability, which resulted in the following distinct periods: (a) grazing autumn temperate pastures from day 1 to 87 (Pastures, 7.37% CP, 68.78% NDF, 52.01% DOMD; 16 April to 12 July); (b) grazing oat crops from day 88 to 121 (OC, *Avena sativa*; 10.83% CP, 34.60% NDF, 80.50% DOMD; 13 July to 16 August); (c) grazing winter pastures with concentrate supplementation from day 122 to 147 (P + C, 11.42% CP, 49.60% NDF, 68.45% DOMD; 17 August to 12 September); (d) fed lucerne hay from day 148 to 180 (LH, *Medicago sativa*; 5.5 kg/hd per day, 21.80% CP, 32.65% NDF, 67.15% DOMD; 13 September to 17 October); (e) fed oaten hay from day 181 to 220 (OH, *Avena sativa*; 7.6 kg/hd per day, 7.38% CP, 63.22% NDF, 56.00% DOMD; 18 October to 22 November). The predominant forage species of ‘Pastures’ and ‘P + C’ were perennial ryegrass (*Lolium perenne*), fescue (*Festuca arundinacea*), white clover (*Trifolium repens*), cocksfoot (*Dactylis glomerata*), chicory (*Chicorium intybus*), kangaroo grass (*Themeda triandra* Forsk syn australis), paspalum (*Paspalum dilatatum* Poir.), purple pigeon grass (*Setaria incrassate* cv. Inverell), setaria (*Setaria sphacelata* var. seric), and rhodes grass (*Chloris gayana* Kunth). Fescue, cocksfoot, paspalum, and rhodes grass were the most predominant species.

Under grazing conditions, animals were moved to a fresh paddock when forage availability to the base of 5 cm from the ground level was approximately 1000 and 750 kgDM/ha for pastures (P, P + C) and OC paddocks, respectively. Detailed information regarding pasture measurements are provided in Imaz et al. [12]. Briefly, forage quantity was measured using an electronic plate meter (EC20, NZ Agriworks Ltd., Feilding, New Zealand) which was previously calibrated based on the type of forages measured. Square bales of hay were offered each week on Monday, Wednesday, and Friday (LH = 350 kg/bale and OH = 425 kg/bale). Hay bales were first placed in a fresh pasture paddock, where animals remained until the end of the experiment, when forage availability to the base of 5 cm from the ground level was 1130 and 830 kgDM/ha at the beginning of LH and OH, respectively. Hay consumption and wastage was estimated by weighing six individual bales of each hay before delivery and then weighing the remaining hay, immediately before offering a new bale. Dry supplementation (C) was offered on Monday, Wednesday, and Friday in a separate feedbunk during period P + C at a rate of 1.25 kg/hd per day by mixing pellets (16.2% CP; 35.3% NDF; 68.7% DOMD) and chopped lucerne-chaff (15.8% CP; 41.1% NDF; 54% DOMD) mixed in a proportion of 75:25. Chemical composition analyses of forages offered, concentrate and MLB was performed following the procedures reported in a companion paper by Imaz et al. [12].

### 2.4. Statistical Analysis

Liveweight data, recorded by the WOW, were filtered for outlying data using methods described by González et al. [13]. Growth rate (i.e., LWC, g/hd per day) was calculated as the first derivative of each predicted LW over the entire period. Then, LW data were averaged by date for each animal if more than one measurement per day and animal existed. The total number of single visits recorded by the EF (n = 3790; including those records without EID and negative records) were analysed following the next steps: (1) Records without an EID were deleted (n = 457); (2) Records with negative intakes were deleted (n = 652); (3) The resulting feeding records (n = 2681), a correlation model between time (sec) and MLB intake (g) was fitted and studentized residuals calculated for each feeding event; (4) Records with residuals lower than −3 or higher than 3 were deleted (n = 20; adding up to 69.52 kg of MLB). Feeding records (i.e., single visit of an animal to the EF) in the final dataset had a mean ± SD of 0.169 ± 0.213 kg/visit whereas feeding records that were deleted had 3.47 ± 2.17 kg/visit.

Supplement intake per visit (g/visit), number of visits per day and visit duration (min/visit) were added up by date for each animal to obtain daily values (daily MLB intake, g/hd per day; daily feeding frequency, visits/hd per day; daily feeding duration, min/hd per day). Visit size (g/visit), visit length (min/visit) and feeding rate (g/min) were calculated from daily values. Data from WOW was also used to calculate the attendance to the water point (WOW attendance, visits/hd per day).

Liveweight and MLB intake data were also used to calculate average values for each animal throughout the entire trial period and a Pearson correlation analysis was performed. Coefficient of variation (CV) between animals and coefficient of determination (R^2^) between MLB intake and feeding behaviour for each feed type were calculated (i.e., for each period with different basal diet). A value of zero was used for those animals which did not consume supplement for the variables MLB intake, feeding frequency, and feeding duration over the entire trial, or in a particular feed type. Additionally, CV between animals and R^2^ between MLB intake and feeding behaviour were calculated for each of three different sub-periods within the feed type ‘Pastures’ (P1 = from day 1 to 30; P2 = from day 31 to 60; and P3 = from day 61 to 87). This was done to compare the repeatability of such measures within the same type of feed (Pastures). Molasses-lick-block intake and feeding duration were transformed to log_10_ to normalize the data before statistical analysis. The intake of MLB was analysed using a mixed-effects linear regression model including feed type and animal sex (Sex) as fixed factors and its interaction. In addition, for the relationship between MLB intake and feeding duration was analysed using a mixed-effects linear regression model with feed type as fixed factor and feeding duration as covariate. Statistical significance was declared at *p <* 0.05.

The relationship between LWC and MLB intake for each feed type was analysed using analysis of covariance. Firstly, LWC and MLB intake for each individual animal were averaged by feed type. These data were then analysed using a mixed-effects linear regression model including feed type and animal sex (Sex) as fixed factors, and MLB intake as a covariate and all possible interactions to investigate linear effects of MLB intake on LWC. Statistical significance was declared at *p <* 0.05. Covariance structure was selected based on the lowest Bayesian Information Criterion. All statistical procedures were done using SAS/STAT software (SAS Institute Inc., Cary, NC, USA).

## 3. Results

Individual MLB intake, feeding frequency, and feeding duration throughout the trial varied among animals (*p <* 0.05; Figure 1) and the latter two variables explained above 85% of the variability in individual MLB intake. Molasses-lick-block intake ranged from 0 to 194.70 g/hd per day (CV = 79.6%; Figure 1a) with similar variability between animals also observed for feeding frequency and feeding duration.

Three animals never registered a visit during the entire trial period (Figure 1b; 11.10% non-consumers) and only nine single records were taken between 20.00 h and 04.00 h. Walk-over-weighing attendance showed a poor correlation with MLB intake (Figure 1d; R^2^ = 0.05) and the lowest CV among animals in comparison with feeding frequency and feeding duration (Figure 1b,c). On average, individual animals attended the water point at least once per day through the entire trial period.

Average MLB intake per feed type was greater under OH compared to the rest of the feed types which did not differ between them (*p* > 0.05; Table 1). Supplement intake was not affected by sex (*p* = 0.11) or its interaction with feed type (*p* > 0.10). The CV between animals ranged from 144.82% for P to 75.25% for OH. The strength of the relationship between MLB intake and feeding frequency or duration changed with feed type. The lowest R^2^ for feeding frequency and duration was observed while grazing OC, and these variables explained over 89% of the variability of MLB intake while animals were consuming P and P + C (Table 1). Within ‘Pastures’, feeding frequency explained 91%, 87%, and 93% of the variability in MLB intake for P1, P2 and P3, respectively (data not shown). Similarly, feeding duration explained 94%, 92% and 94% of the variability in MLB intake for P1, P2 and P3, respectively (data not shown). However, feeding duration explained a higher proportion of the variability in MLB-intake compared to feeding frequency while animals fed on LH and OH. Regression coefficients of MLB intake against feeding duration ranged from 16.53 (OC) to 27.12 (OH) g/hd per min whereas intercepts did not differ from 0 for all feed types.

Table 2 shows the correlations between MLB intake amongst different periods of time or feed types. Results indicate that MLB intake of individual animals in one period was positively correlated with the intake in any other period (*p <* 0.05). However, the strength of the correlation was not consistent across feed types. For example, the amount of MLB consumed while grazing Pastures was moderately correlated (R < 0.60) with the amount consumed in other feed types. In contrast, the correlations between the amount of MLB consumed while on OC, P + C, LH, and OH were high (R~0.81).

Liveweight change of individual animals over the entire period ranged from 416 to 719 g/hd day (Mean ± SE = 553 ± 43 g/hd per day). Additionally, the average LWC for each feed type, when MLB intake is zero, is indicated by the intercept; being P + C and OC the fastest and lowest growth, respectively. Liveweight change was affected by the interaction between feed type x MLB intake (*p <* 0.05). Thus, results are presented for each feed type (Table 3). Intercepts indicate a positive growth of animals across all feed types. The regression coefficients differed (*p <* 0.05) from zero for P + C and LH indicating a positive linear relationship between LWC and MLB intake during these periods. No relationship between LWC and MLB intake was found for the rest of the feed types (Table 3; *p* > 0.05).

Molasses-lick-block intake throughout the entire trial period was positively correlated with feeding frequency, feeding duration, visit length, and visit size (Table 4; *p <* 0.05) but not with feeding rate (*p* > 0.05). However, the strongest correlation coefficient (R > 0.90) was observed for feeding frequency and feeding duration. No correlations between WOW attendance and LWC, MLB intake and feeding variables were observed (Table 4; *p* > 0.05). Liveweight change was positively correlated with MLB intake, feeding frequency, feeding duration, and visit length (Table 4; *p <* 0.05; R~0.45). Initial and final LW were positively correlated with MLB intake but only final LW was correlated with LWC (Table 4; *p <* 0.05).

## 4. Discussion

The aim of the present study was to assess self-fed supplement intake, feeding behaviour, and LW of individual animals to link these outcomes to individual animal daily LWC. In-paddock technologies were used to measure MLB intake and LW because these technologies offer the potential to collect data continuously and remotely [11,12]. Our results showed that these technologies enabled individual assessment of MLB intake and LWC across periods with different feed types and therefore examination of between- and within-animal variation. The MLB intake of individual animals was highly variable over the entire trial. The intake of MLB across the trial influenced LWC and this effect was only found while certain feed types were fed (P + C, LH). Therefore, the combination of WOW and EF technologies allowed establishing correlations between LW and feeding parameters both in the long-term across the entire trial and within shorter periods of time when animals consumed particular feedstuffs. This approach of combining EF and WOW could allow automated monitoring and management systems of grazing beef cattle.

The EF revealed a large variability of MLB intake (CV ranging from 75 to 128) among individual animals over the entire trial. Findings from the present study agree with previous work in beef cattle. For instance, Kendall et al. [7] reported a CV between animals of 57% and 82% for grazing heifers and steers, respectively, consuming MLB offered in individual pens. Graham et al. [6] observed even higher MLB-intake variability between individual steers, ranging from 77 to 488 g/hd per day, estimated by using markers over a six day period. The variability in MLB intake of the present study can be attributed to a wide range from animals which did not consume any supplement (non-consumers, 11%) to animals that consumed 194 g/hd per day. Bowman and Sowell [1] reviewed 15 studies, including beef and sheep data and reported that the percentage of non-consumers varied from 0% to 50% with an average of 14.32%. However, none of these studies measured the amount of MLB intake individually in grazing animals. The most common intake measurements were done by using markers [14], total faecal collection of indoors animals [15], and identifying consumers by colouring them [16] for periods no longer than four weeks. Based on previous research and our results, high MLB intake variability and attendance to the blocks seems a consistent attribute of animals consuming MLB. However, MLB intake and LWC variations observed in different studies proved to be affected by block formulation (e.g., hardness) and composition such as urea content [1,5,10,17] and these factors could modify block intake patterns with potential impacts on performance [1,5]. Therefore, results from the present trial describing variations in MLB intake and LWC cannot be considered a general assumption and every situation may need to be monitored. A companion paper of the present study [12] has also demonstrated that multiple factors such as changes in forage quantity and quality affect the temporal variability in MLB intake and LWC. Therefore, the ability of in-paddock technologies to monitor animals can help understanding the variability in MLB intake and LWC which seems to be driven by complex interactions between the quality and quantity of the main feed available, block composition and formulation, and animal characteristics.

The use of EF could be useful to identify variability in supplement intake among individuals and also to measure feeding behavior of animals. In addition, feeding behaviour could be used to predict MLB intake when measures of MLB disappearance cannot be taken. Both feeding frequency (R~0.93) and duration (R~0.95) showed a strong positive correlation with MLB intake over the entire trial. Furthermore, these associations across the entire trial were similar to those found within sub-periods of the same feed type, i.e., Pastures. However, these relationships can be affected by changes in feeding behaviour and social interactions. Wierenga and Hopster [18] reported that animals interacted with an EF by alternating rewarded (visits leading to supplement consumption) and unrewarded visits based on feed availability. In addition, prediction of MLB intake from feeding time and visit frequency was affected by feed type of the main forage diet in the present study which may limit the application to feed types which were not part of the study. Results also suggest that feeding duration could be a better predictor of MLB intake than feeding frequency under varying feed types. Feeding rate, visit size, and visit length explained lower proportion of the variation compared to daily feeding frequency and duration, and thus these measures are less suitable to predict MLB intake. In addition, block hardness could affect intake [10] and also affect the relationship between feeding behaviour and intake so these prediction equations are not universally applicable. Nevertheless, data on feeding behaviour could be useful to estimate supplement intake of individual animals and allow for the development of low-cost monitoring systems based on EID readers, without the need for weighing cells to measure feed disappearance.

The lack of simultaneous supplement intake and LW data has limited the study of cattle growth responses to molasses-based supplements [5]. Beef producers often feed a wide range of forages throughout seasons and the present study showed that associations between MLB intake and LWC between animals changed according to feed type. However, further research is needed to confirm the results obtained in the present study using a higher number of animals and manipulating the quantity and quality of the basal diet over the time. The MLB used in the present study was formulated for year-round supplementation providing both bypass protein (from cottonseed meal) and non-protein N (3% urea). Remote monitoring technologies helped to quantify the variability in performance across individual animals attributed to MLB intake during both the entire trial and while offering particular feed types. This was confirmed by the positive linear relationship between MLB intake and LWC only being evident while feeding P + C and LH. However, there were other factors not considered in the present study that could strongly influence individual LWC such as total individual feed intake (feed + MLB) and social dominance. Bowman et al. [5] reported that LWC was mostly improved when feeding molasses with a high proportion of non-protein N (>10%) and low-quality forages. However, positive responses were also reported when the CP of the diet was increased [19]. In addition, across the entire period of the present trial, MLB intake was positively correlated with LW which could indicate that heavier animals were dominant at the feeder. Furthermore, the coefficient of correlation between MLB intake, expressed as percentage of body weight, and the LW of animals over the entire trial was 0.42 (*p <* 0.05, data not shown). These results confirm that heavier animals ate more MLB however the reason for these findings cannot be confirmed in the present trial. Social dominance may have played a role in these findings despite the fact that the EF was used for only 77.5 min/d and thus the feeder was unoccupied for the majority of the day to allow subordinate animals to consume MLB.

Integrating both EF and WOW data streams could enhance the study of factors involved on animals’ motivation to consume self-fed supplements. For example, MLB intake varied greatly amongst individual animals and feed types, and feeding frequency was lower than 1 visit/hd per day, even for frequent consumers. The feeding behaviour of MLB supplements does not seem to be driven by satiety and hunger mechanisms as reported in previous studies [20] where total mixed rations (TMR) and loose supplements were fed to predict the probability of the animal beginning a meal, which increases with the time since the last meal. However, we did not find any evidence of such mechanisms in the present study (data not shown). Interestingly, WOW attendance was not associated with feeding behaviour, MLB intake, or LWC in the present study. In addition, cattle attendance to the water point was less variable than visit frequency of MLB suggesting that WOW attendance may be strongly influenced by the gregarious behaviour of the herd. Thus, WOW attendance may reflect the herd’s time of drinking preference when most animals of the herd go to the water point but not necessarily consume supplement. Further studies should be conducted to elucidate factors driving individual intake of self-fed animals which may offer potential to improve animal performance by reducing the variability of intake amongst animals.

## 5. Conclusions

The combined use of in-paddock technologies, such as electronic feeders and automatic weighing scales, allow to continuously monitor performance of individual animals. The analysis of data obtained from these technologies revealed associations between molasses-lick-block intake, feeding behavior and growth rate in real-time which could be useful to quantify the impacts of MLB supplementation across and within different types of feed. The ability to capture these associations could improve nutritional management and supplement formulation and tailor these to the forage being consumed. Finally, the findings of the present study could contribute to the future automation of farming activities, such as timing, type, and amount of supplementation tailored to the requirements of individual grazing beef cattle.

## Figures and Tables

**Figure 1 animals-10-00093-f001:**
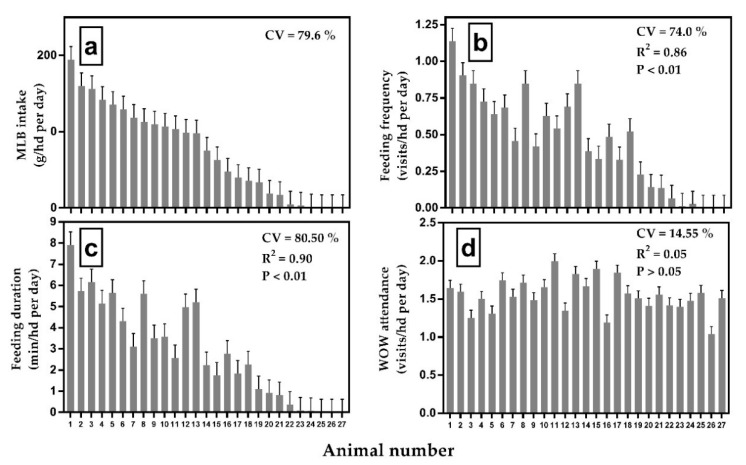
Molasses-lick-block (MLB) intake (**a**), feeding frequency (**b**), feeding duration (**c**), and walk-over-weighing (WOW) attendance (**d**) of individual animals (Mean ± SE) during a 220 day MLB supplementation trial. Animals (columns) are ordered on the x-axis in decreasing amount of MLB intake in all panels. Coefficient of determination (R^2^) was calculated for each variable against MLB intake. Coefficient of variation (CV) was calculated among individual animals.

**Table 1 animals-10-00093-t001:** Molasses-lick-block (MLB) intake, coefficient of variation (CV) of MLB intake between individual animals, and coefficient of determination (R^2^) to predict MLB intake from feeding frequency and feeding duration for each feed type.

Feed Type *	MLB Intake	CV	Frequency	Duration (min/hd per day)			
	(g/hd per day)	(%)	R^2^	R^2^	β	SE	*p*-Value
**Pastures**	44 ± 11.6 b	144.8	0.93	0.96	21.4	2.11	<0.001
**OC**	42 ± 14.7 b	99.5	0.66	0.66	16.5	3.12	<0.001
**P + C**	62 ± 15.7 b	128.1	0.89	0.94	23.2	2.39	<0.001
**LH**	71 ± 15.0 b	102.0	0.56	0.80	21.1	1.94	<0.001
**OH**	173 ± 14.8 a	75.25	0.72	0.83	27.1	1.36	<0.001

***** Pastures, grazing autumn temperate pastures; OC, grazing oat crops; P + C, grazing winter pastures with concentrate supplementation; LH, lucerne hay; OH, oaten hay.

**Table 2 animals-10-00093-t002:** Pearson’s correlation matrix between average molasses-lick-block intake (MLB, g/hd per day) of individual animals consuming different feed types. Values above the diagonal are correlation coefficients, *p*-values are below the diagonal.

Feed Type *	Pastures	OC	P + C	LH	OH
**Pastures**	1	0.521	0.470	0.584	0.536
**OC**	<0.01	1	0.882	0.825	0.805
**P + C**	0.013	<0.01	1	0.769	0.770
**LH**	<0.01	<0.01	<0.01	1	0.830
**OH**	<0.01	<0.01	<0.01	<0.01	1

*****—Pastures, grazing autumn temperate pastures; OC, grazing oat crops; P + C, grazing winter pastures with concentrate supplementation; LH, lucerne hay; OH, oaten hay.

**Table 3 animals-10-00093-t003:** Prediction equations for liveweight change from molasses-lick-block intake (MLB) of weaner cattle consuming different feed types. Regression coefficient (β), intercept (α), and *p*-value for the intercept and regression coefficient.

Intercept				Linear Regression Coefficient		
Feed Type *	α	SE	*p* Value	β	SE	*p*-Value
**Pastures**	236	42.1	<0.01	0.350	0.550	0.520
**OC**	180	49.4	<0.01	0.500	0.840	0.540
**P + C**	1037	44.1	<0.01	0.980	0.440	0.020
**LH**	790	48.7	<0.01	1.580	0.480	<0.01
**OH**	804	58.6	<0.01	−0.028	0.274	0.910

***** Pastures, grazing autumn temperate pastures; OC, grazing oat crops; P + C, grazing winter pastures with concentrate supplementation; LH, lucerne hay; OH, oaten hay.

**Table 4 animals-10-00093-t004:** Pearson’s correlation matrix between molasses-lick-block intake (MLB, g/hd per day), feeding frequency (visits/hd per day), feeding duration (min/hd per day), liveweight change (LWC, g/hd per day), visit size (g/visit), visit length (min/visit), feeding rate (g/min), walk-over-weighing attendance (WOW; visits/hd per day), and initial and final liveweight (LW; kg/hd) during a 220 day MLB supplementation trial. Values above the diagonal are correlation coefficients, *p*-values are below the diagonal.

Items	MLB	Frequency	Duration	LWC	Visit	Visit	Feeding	WOW	Initial	Final
	Intake				Size	Length	Rate	Attendance	LW	LW
**MLB intake**	1	0.928	0.950	0.444	0.479	0.700	0.111	0.217	0.461	0.598
**Frequency**	<0.01	1	0.975	0.415	0.190	0.591	−0.033	0.257	0.366	0.502
**Duration**	<0.01	<0.01	1	0.447	0.280	0.722	−0.076	0.133	0.369	0.543
**LWC**	0.021	0.032	0.020	1	0.283	0.413	0.390	−0.012	0.008	0.451
**Visit size**	0.018	0.374	0.186	0.180	1	0.650	0.565	0.011	0.145	0.250
**Visit length**	<0.01	<0.01	<0.01	0.045	<0.01	1	0.065	−0.271	0.116	0.351
**Feeding rate**	0.605	0.878	0.723	0.060	0.004	0.762	1	0.321	0.019	0.114
**WOW attendance**	0.277	0.195	0.507	0.954	0.958	0.200	0.126	1	0.201	0.068
**Initial LW**	0.015	0.060	0.058	0.969	0.498	0.588	0.929	0.315	1	0.906
**Final LW**	<0.01	<0.01	<0.01	0.021	0.251	0.101	0.603	0.742	<0.01	1

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
