# Peer review of "Application of In-Paddock Technologies to Monitor Individual Self-Fed Supplement Intake and Liveweight in Beef Cattle"

_animals, 2020, doi:10.3390/ani10010093_

Round 1
Reviewer 1 Report
The present manuscript by Imaz et. al., evaluate the use of ‘in-paddock’ liveweight (LW) and electronic feeders (EF) as tools to study the relationship between intake of molasses-lick blocks (MLB) and growth of individual animals. The authors concluded that these technologies enabled individual assessment of MLB intake and LW in order to examine variation between and within animals. Furthermore, the present study indicates an association between MLB intake, feeding behavior, and growth rate. This reviewer acknowledges the authors for the important effort of trying to increase the body of knowledge on the currently available technologies for pasture-based feed intake and body weight measurement. However, there are concerns with the methodology and description of the present study. These concerns are listed below:
General comments
-Has the system utilized to monitor feed intake been previously validated for measuring intake of molasses blocks? One of the main concerns with the present study is whether the estimated intake by these systems are comparable with manual measurements of intake in the context of blocks.
-Another major concern is the description of the statistical analysis. The description of the statistical needs considerable editing. The authors should keep in mind that the statistical analysis must have enough details so that the reader can understand exactly each component of the statistical model utilized. The current description of the statistical analysis makes the understanding of the results difficult for the reader and impacts the interpretation of the manuscript. Also, it is recommended that the authors include the rational of a given statistical approach when possible.
Specific comments
Line 31 – Describe “feed types”.
Line 61 – Substitute “could be used…” by “might have the potential….”. Alternatively, authors could add a citation that corroborates with this statement.
Line 67 – Substitute “reveals” by “indicated”.
Line 70 – Describe abbreviation (LWC) at first use.
Line 83 – Replace “weaner” by “weaned”. More information is needed regarding animals origin (same location or different locations), breed and how was that accounted for in the present study.
Line 83 – “Cattle were tagged with”
Line 85 – “AND grazed”
Line 86 – Mention that details regarding the composition of pastures and nutritional value of supplements are reported later on the manuscript.
Line 100 – 0.8 m width
Lines 108-109 – Change sentence to “Animals that were not supplemented were grazing together with ….”
Lines 119-124 – Add a space in between % and NDF, CP, DOMD, etc. Also, it is not clear if the nutritive values reported in the text are associated with the forage or supplement. Make sure this is clear for the reader. What do you mean by autumn temperate pastures? Grazing winter pastures?
Lines 125-127 – How was forage availability quantified?
Line 142 – What do the authors mean by “first derivative of the predicted live weight”. Why not simply report average daily gain as measure of growth rate? Include rationale in the manuscript.
Line 147 – What do the authors mean by feeding records. Average amount of feed consumed per day? If yes, add kg/had/day as unit.
Lines 153-155 – What do the authors mean by data? The response variables? This portion of the statistical analysis is confusing. It is recommended that the authors properly state the rationale for their statistical models and list their response variables in a manner that facilitates the understanding of the analysis. The reader should be able to replicate the statistical model utilized in the present experiment based on the description of the statistical analysis.
Line 159 – Include rationale for log-transforming the data. Also, what do you mean by feed type? The different periods?
Line 163-167 – Did the model include main effects of feed type and animal sex? It is worded as if only the interaction was included in the model. What was the rationale for including MLB as covariate?
Line 177 – Figure 1, Panel D.
Line 212 – Add line below the table headings (Feed type, pastures, OC, P+C, LH, OH). See table 3 as an example.
Line 219 – Report r-squared in order to explain the proportion of the variation explained by the proposed regression.
Line 236 – Provide brief description of the table in the table’s title. Thoroughly described table contents on the table’s captions (bottom). Use superscript to refer to specific components of the table.
Line 250 – this effect was found ONLY when animals were exposed to specific feeding conditions (P+C, LH)….
Line 263 – spell out percentage
Line 285 – briefly guide the reader through what “rewarded” and “unrewarded” means.
Reviewer 2 Report
General comments
This paper has some interesting insights, in particular that the combination of walk-over-weigher and Electronic feeder can work well to characterize feeding behavior of grazing animals (of supplemented animals).
Whilst the title of the paper appears to focus upon the technique used, much of the paper focuses upon variability between the feeding treatment periods. The degree of variation between animals during different feeding treatment phases is highlighted through regressions and correlations. But the data as presented in the body of the paper does not really match up statement from the Simple Summary “ …with animals being regular consumers while on one feed type but occasional consumers with other feed types”. Similar, less descriptive wording appears in the Discussion re variation in between-animal intake between background diets periods. The paper should make less focus upon comparing the basal feed treatments.
In that respect, it can be argued that the study confounds the issues of dietary treatments, which is just a sequence of difference base forages over several months, with time itself. There is no proof/evidence provided that the differences described between different baseline diets/forage types is not merely an effect of the passage of time, and animal, and inter-animal, changes occurring. There is/are no adequate control(s) - just a time series of different treatments. Too much should not be made of feeding behaviour differences between different cattle being linked to the different basal forage types. There is variation in mean intake per head between different baseline feed treatments (LH versus others) and I cannot see an animal-based hypothesis as to why. These are fundamental weaknesses of the paper. As an example, at Paragraph line 204+ and Table 2 , the argument is made that these show inconsistency between treatment periods. My reading of this data, which all have positive correlations is that it shows consistency across periods, It is inevitable, given basic biological variation across time that these correlations are not perfect.
Variability between animals in different dietary periods (and as a way to just look at time, the 3 sub-periods in the 80 day long P period) could be shown by histograms of intake per animal across all periods. The variability would then be visibly shown.
A further way to look at the long Pasture treatment period of almost 90 days, and sub-divide this to provide a point of comparison with the shorter c 30 days periods on the other basal diets/treatments.
The paper suggests that the different background diets had some effects, in part by pointing to significant regressions for some periods and not for others. But this falls a little into the trap of suggesting that no, or low levels, of statistical significance prove that there is the absence of an effect. Whilst of course, it may be that the study has too little power, or there is experimental noise (and the study had only 27 animals and treatment periods of less than 30 days).
In terms of the key aim of the paper, the accurate measurement of individual intake, per meal, and in total, I believe it is important that have statement on the comparison between the intakes accumulated from individual intakes by the method used, and the gross amounts offered/disappeared.
So my key suggestion is that the paper reduces focus on between treatment differences linked to the diet/nutrition, but just showing variability against time/a diversity of changing nutritional background. Emphasis should be more on showing that the combined techniques worked well and that relationships between supplement feeding behaviour measures and between these and LWC itself were evident, even when basal diets, intake levels of supplements and growth rates varied. So, not on saying the different diets themselves had impacts, they were just a convenient way of creating diversity in the data sets. Without proof, and a biological hypothesis, for the role of background diet, then would suggest this is downplayed.
There are some clarity issues in the methods; how many non-supplemented animals were there; how were zero intake animals dealt with in various statistics; what were the totals of missing data removed, and any mismatch between bottom-up calculation of intakes from the ‘cleaned’ data and the top down amount of MLB offered (this would provide evidence of accuracy of methodology); was there evidence of social dominance having no, or a real, effect? A line drawing of the layout would help to explain how the set up worked.
Finally throughout, there are quite a lot of errors, and clumsiness in English. Some of this leads to uncertainty in meaning, others times it is just irritating. Many are pointed out in detailed comments below, but many more not covered. A thorough proof read by a fluent English-writer is recommended before further submissions.
Detailed comments by line number
29 WOW this is conventionally a Walk-Over-Weigher , here the acronym from in-paddock weigh scale …suggest ‘in-paddock walk over weighscale (WOW)’ in full, before acronym use..
32…unnecessary precision…194 g is sufficient..
43 sentence a bit clunky, as first sentence make it clearer..and simplify into two.
45 labour constraintS (Note plural)..sentence poorly written.
55 ‘tech-based’ I know this sort of terminology is moving into written texts but ‘technology based’ better.
59 only A few…
59 hardness not hardiness
75 …measurement of….rather than measuring…this whole (important) sentence reads poorly and would be better re-phrased
88 put details in brackets earlier in sentence next to feed, or provide a separate sentence describing block.
89+ chemical composition listing looks odd..check formatting with journal requirements
98 provide details of equipment in brackets next to mention of weigher etc..
108...’No supplemented’ is incorrect grammar, use either ‘none-supplemented’ or write in longer phrase ..’cattle allocated to a no supplement…were..’ etc.. state number here, or earlier when n=27 is mentioned. It might not be relevant but whether the 27 experimental study animals were with 5 or 50 none-supplemented might have some relevance.
117 forage offered here per head …so relevance of size of whole herd, how 27 selected, ‘were offered..’ would be better terminology..
117+ this data would be best within a table as it provides the baseline treatments, their periodicity and the chemical analyses. Add the number of days for each treatment period.
121 ..how and when are supplements offered in the P+C treatment period? The chemical analysis presumably refers to the concentrate in ‘C’ but what about the analysis of ‘P’ during these periods, it would likely be different to the analysis in period P
126 say a bit more of how these values were measured/estimated ..if standard Australian technique then need a little more info..
140 provide name(s) of authors and then [16]
142 as multiple LWC periods, the whole period and each treatment period, then ‘....of each predicted liveweight curve for the different treatments periods’ or similar
143 omit ‘Total’ – it all the MLB data not just the Totals?
145 ‘..was fitted..’
145 it is not clear for what period the regression, of time(sec) and intake, was taken. Was it daily, weekly, treatment period or whole study?
146 ‘studentised’ ?? please write more formally
146 ‘3’ is this 3g?
146 “Records with residuals lower or higher than 3 were deleted” . I’m missing something here, or this means that all records were deleted as all data must have had residuals either lower or higher than 3…I am guessing it might be outside the range of -3 or +3. Please review this terminology.
149 “in the final dataset had a mean ± SD of 0.169 ± 0.213 kg while deleted feeding events had 3.47 ± 2.17” some explanation of what/why feed weights during deleted feed events were so high …? What types of error were involved?
Whilst proportion of data deleted was noted for steps 1) and 2) there is no similar statistics for steps 3) and 4). Please include to be consistent and to identify how much data deleted in totals. There are obvious sources of error – where animal visited but didn’t eat, where animal visited and ate but the actual weight of MLB was not included in data set (the +/- 3 residual?) . It would be useful, even essential, to report the proportion of MLB that disappeared compared to the weight that was allocated to individual intakes. i.e. comparing the totals from individual intakes with the totals coming from weighing in and out of the feed …was it trivial, or considerable? This type of information is need for fulfilment of hypothesis on accuracy i.e. that the methods can measure intake accurately, and ascribe reasonably accurately to individuals.
Was error /deleted data evenly spread across animal ID, or were there any patterns? Was it animal based or random?
174 this section could be clearer with a line drawing/plan of the set-up.
Methods section. More information is needed about animals with zero or very trivial intakes. The short summary makes mention of these animals. It is not obvious whether all, and only, the same 3 cattle, had zero intakes during each, and all, treatment periods. Whilst, they could/should not be used for correlations for feeding behaviour data as no data, I presume that LWC vs intake could include. Please be clear throughout.
Table 1 ..186… (Zero values as above) were the 3 animals that never visited, or indeed other animals that never visited during the discrete ‘different forage’ periods, with zero intakes included or excluded here for mean values. Not clear throughout whether means include zero animals, including table 1 , column 1 MLB intake g/hd/day was hd=27 or a lower number?
Table 1 a and b subscripts (check how journal wants these shown in tables – presume b (Oaten) is significantly different from rest. Table footnote should say so
Table 2 three decimal places of R2 seem unjustified – check journal policy.
204 ‘..while consuming 2 different feeds…’ . this is poor English and clarity. The correlations statistics are between MLB intake between periods of time when animals were on different feeds.
205 ‘was not consistent’ across feed types/periods. I believe the contention that MLB intake was not consistent is unproven by Table 2. Indeed I think this suggests consistency – they are all positive correlations, even though some have higher levels of agreement than others. Where is the evidence that they are not broadly consistent in this Table?
The results/discussion/conclusion suggest variation between diet periods in terms of which animals ate most/least, their rank order (correlations being better at seeing consistent/inconsistent rank orders than 1:1
A fundamental issue is that there is little proof/evidence that the base diet had an impact upon feeding behaviour, rather than might be seen in a time series. Treatment OC had higher mean intakes of MLB, but otherwise the variablility in MLB between animals, driven by the base diet is not proven. How might it be?
I would query whether the long duration treat period (P) would show within period/treatment variation similar to that for the later, short interval periods/treatments. If you were to split period 1 P which is 87 days into 3 – and do within-treatment periods comparisons for say P (day 1 to say d30) vs P(31-60) vs P(61-87) would this show different characteristics to those looking at P vs OC vs OH etc? Until this type of evaluation is carried out, then I believe there is a confounding effect between the basal diet type and the timeline. There are many ways that between-animal intakes may vary over time and the conclusions here that this is linked to diet type is just not proven.
216-17 it is unclear how LWC over the whole period could be affected by ‘’interaction of MLB intake and Feed type”. Each animal had a single LWC for the whole period and all animals had the same pattern of treatments, so this sentence/section, surely, cannot be correct.
Table 4 same comment re 3 decimal places. And consistent approach re p, sometimes shown as p<0.05, sometimes P=0.001 .
It is difficult to link the Correlation values and p values with such a large table. Combining the two via Bold, ordinary, italics etc or stars with just the upper side of the matrix.
The order of factors in the table is in a completely different order to the Table legend/title. It would be better if they were the same.
Tab 4 – spelling ‘length’ x 2
Discussion
243 Can’t look at LWC (part of list) to look at LWC..Re-write this sentence ..look at inter-relationships between and identify which factors related to LWC..??
244 – why is near real-time so important…another sentence noting that logical extension is that management decisions can be made?? Suggest spell it out…paper needs to be clear whether the scope of paper is to look at factors scientifically, or find management opportunities from data flowing from this type of equipment and data capture (or both). It is clear that that the equipment performed, what would be the management benefits? What actions would be taken? For example, removal of non-feeders for training and return?
249 this sentence isn’t entirely clear – the LWC for each treatment period aren’t shown. They could be shown in table 2 etc.. the regressions between MLB and LWC for each period are shown, but not the mean values of LWC and without these it is difficult to consider the biological interactions. The intake of supplement during the OC period is high, but did this offset/support continuing positive growth, or was the period characterised by low growth. The negative slope of MLB intake and LWC suggests something different is happening during this phase.
250 ‘Therefore… ‘ this sentence doesn’t flow well from previous sentence and English grammar should be corrected
Paragraph 1 of Discussion is unconvincing – part summary, part conclusion
262 unnecessary decimal points..
263 use ‘percentage’ not ‘the %’
269 ‘showed’ grammar error ..hardness not hardiness
271 this mini-conclusion is a bit odd here, and it is not clear why/what is meant by it…..does MLB feeding behaviour influence LWC or not?
273 impact of LW on intake – was this greater than metabolic weight – i.e. non-proportional? Where in the results is this factor shown/described?
282 “Feeding frequency and duration showed a strong positive 282 correlation with individual MLB intake “ [check what results show…it is each, or in combination??] Clarity needed.
283 this sentence needs linking better to previous and next sections
291-295 two sentences don’t fit well together, first saying ’prediction equations not universally applicable’ which is clear, second then says “Feeding behaviour can be useful to predict MLB intake of individual animals” . corrections to grammar needed too . ‘Allow developing’ is not good English.
“Nevertheless data on feeding behaviour could be useful to estimate supplement intake of individual animals and allow the development of low-cost monitoring systems based on EID readers, without the need for weigh cells to measure feed disappearance” and finish there without repetition and too narrow a focus upon MLB…
291 hardiness wrong again
296 this sentence and next sentence could be written clearer. The reference [5] is in an odd place, at the end of sentence would seem better if this whole statement is a conclusion/observation by Bowman. If it is instead an inference by current authors then this needs to be made clear. The ‘for long periods of time’ and ‘throughout the year’ statements are somewhat overlapping and could be best just left out, as the next sentence says it all in reference to farming systems when basal diets change in either or both diet quality and type.
299 ‘….that different responses to MLB are expected between animals and forages as well’ this needs writing clearer and in relation to evidence provided. The ‘as well’ is not needed/meaningless. I think this sentence is saying that there are ‘variations in animal intake and growth responses between animals and these responses are different between different diet types’. I dispute whether the different diet type effects is proven or explained.
307 social dominance is mentioned here for first time. Unclear whether authors considered that the EF/WoW setup may have enabled or mitigated against social impacts. Heavier animals had higher MLB intake, though it is unclear whether this effect was in line with metabolic demand or was greater than predicted by body weight differences. This needs calculating and pointing out, as it may be important to the interpretation of results. Intakes higher/lower for heavy/light animals than predicted by nutritional requirements might suggest an impact of dominance, or the absence of this effect the absence of bullying/social dominance
310 makes this point, without any evidence or discussion. Correlation cannot provide evidence of cause and effect, but if larger animals having a higher intake of MLB than the metabolic body weight predicts could be helpful here.
318 poor phrasing ‘….probability to begin a meal’ better written as’…. the probability of the animal beginning a meal’
‘This probability increases with the time since the last meal and fits mixed distributions with probability density models from at least two populations of the length of intervals between successive feeding events’ This sentence should be re-written making it clear that ‘Tolkamp et al found/said/noted that ….’ The main element of this sentence, though, does not appear to make any sense.
It isn’t clear what analysis was done by current authors in relation to meal periodicity and probability, but a typically 1 or less supplement visits per day, then it is clear that supplement use was not characterised by typical meal patterns in cattle.
Best thing to maybe say is that ‘Tolkamp showed x, we looked in our data for y, but no effects were found. The ‘data not shown’ statement is then not needed.
No discussion on MLB intakes (average and variability) and the quality of the main diets…did it differ due to diet type and quality or not. This fundamental point needs covering. As LWC and MLB overall, and for 2 of the 5 diet periods was positively related to each other, but much is made of variation in overall mean intakes, and slopes/intercepts/significance of regressions, then was there evidence that overall intake of MLB was influenced by baseline diet quality?
Whilst some mention of value of this type of instrumentation, especially real-time, no real discussion of practical application of this type of data. What type of management intervention could be made and what is its potential? Non-feeders are not discussed here, though low intakes are mentioned in the Simple Summarry, its not clear whether zero intake data is included, and what impact upon performance is found, and whether none- or infrequent and irregular feeders and low quantity feeders could be managed differently..
334 better ‘…could be useful..’
335 ‘these correlations’ .better wording would be ‘relationships’ avoiding the statistical connections and picking up the biological relationships
The companion paper is mentioned at least twice, but it is unclear how it differs and fits in.
Round 2
Reviewer 1 Report
General comments
Reviewer 1 = Has the system utilized to monitor feed intake been previously validated for measuring intake of molasses blocks? One of the main concerns with the present study is whether the estimated intake by these systems are comparable with manual measurements of intake in the context of blocks.Authors = the same electronic feeder has been used before to measure the disappearance of supplements of grazing animals (Reuter et al., 2017, Williams et al., 2018). As far as we know, there is no study comparing molasses-block disappearance measured manually or using electronic feeders. However, the technology was acceptable to perform individual measures because of the following reasons: a) It has been able to detect feeding events with small quantities of supplement consumed in previous studies (< 15 g/visit; Reuter et al., 2017, Williams et al., 2018); b) we did calibrate and monitored EF performance, on average, twice a week manually (weighing blocks and re-calibrating the system); c) we were able to detect unnormal weights (e.g. a single visit > 5 kg/hd; animals hitting the bin,) through the remote monitoring systems; d) we recorded feeding events using video cameras to make sure that negative and too high values were due to animals hitting the bin as reported in previous studies (Reuter et al., 2017) (lines 116 – 119).
-Include the information included in the above mentioned bullet points (a, b and c) in the materials and methods of the manuscript.
Specific comments
10) Reviewer 1 = Lines 119 -124 – Add a space in between % and NDF, CP, DOMD, etc. Also, it is not clear if the nutritive values reported in the text are associated with the forage or supplement. Make sure this is clear for the reader. What do you mean by autumn temperate pastures? Grazing winter pastures?
Authors = comment addressed in the new version of the manuscript. These were the same group of paddocks. Autumn and winter indicate the season of the year when these paddocks were grazed.
Indicate what kind of forage the animals were grazing in each season in the manuscript (include forage species)
18) Reviewer 1 = Line 212 – Add line below the table headings (Feed type, pastures, OC, P+C, LH, OH). See table 3 as an example.
Authors = comment addressed in the new version of the manuscript.
Do it for table 2
Reviewer 2 Report
This paper is much improved in terms of general readability throughout. The authors have dealt with all of the points raised in Review 1.
There are a few issues that I still think need dealing with. I have referred to original comment number of new line numbers of re-submitted manuscript.
Line 29 …systems…, should be ..system…?
Line 44 ..has been challenged.’ Is this better as ‘has been challenging’ or ‘is challenging’ ?
Comment 5) Authors have not really taken my main point about the general accuracy of the EF data. The author’s added some comment at new lines 116+ that refer to checking the accuracy of the weighing equipment. These comments are useful. But what I was looking for was some statement with some weight values about the accumulated totals of individual intakes, as measured from the ‘cleaned’ data for individual offtakes. This would then be compared with gross amount of feed that was offered/disappeared. For example, “ Over the period of the study a total of 145kg of MB was consumed (weight of block offered minus any remainder weighbacks) compared with 153 kg in total measured into individual animal accounts” [ these numbers are just illustrative]. This would indicate that the basic methodology, of both the weighing accuracies and the data cleaning, is sound. If however, the numbers were 145kg and 250 kg respectively it would indicate that some other fundamental errors are unaccounted for. Please add a statement on the comparison of ‘top down’ and ‘bottom up’ measurement of total feed intake and then if need be include in Discussion/Conclusions. This is an important indicator of the reliability of the EF methodology, and subsequent cleaning of data.
Line 146 – why is the list of authors in full - should be ‘et al’?
L 142 – similar – 3 authors this time
Comment 7) ‘line drawing’ . Now that Imaz et al 2019 is published then it can be referred to allow readers to see the plan of the WoW/EF setup. Would suggest this is done at new line 110 or thereabouts specifically – A figure showing the layout of the…is given in Imaz et al…[13] and …. The companion paper story is mentioned on a number of occasions, but the layout of the WoW/EF would be useful but earlier in manuscript than currently
Comment 20) . I asked for some methodology of how 1000 and 750 kg/ha of sward abundance was measured/estimated. But new lines 130-132 gives no change to the version 1. I would suggest more methodology such as ‘More information on pasture measurements are provided in Imaz et al [13], and maybe including as per that paper “Forage quantity was measured using an electronic plate meter (EC20, NZ Agriworks Ltd, Feilding, New Zealand).” . I appreciate that the actual moves of cattle may be based on ‘by eye’ estimates, but these would be based on actual measurements using the platemeter. This section just needs to provide some concise methodology as to how 1000 and 750 were assessed.
Line 235 onwards. Here mainly Table 3 is being referred to. It might be better to introduce ‘table 3’ earlier in this paragraph than it is.
